# Structured Bayesian Meta-Learning for Data-Efficient Visual-Tactile Model Estimation

**Shaoxiong Yao[1], Yifan Zhu[2], and Kris Hauser[1]**

[1]Department of Computer Science, University of Illinois at Urbana-Champaign, IL, USA.
[2]Department of Mechanical Engineering and Materials Science, Yale University, CT, USA.
{syao16,kkhauser}@illinois.edu, yifan.zhu@yale.edu

**Abstract:** Estimating visual-tactile models of deformable objects is challenging because vision suffers from occlusion, while touch data is sparse and noisy. We propose a novel data-efficient method for dense heterogeneous model estimation by leveraging experience from diverse training objects. The method is based on Bayesian Meta-Learning (BML), which can mitigate overfitting high-capacity visual-tactile models by meta-learning an informed prior and naturally achieves few-shot online estimation via posterior estimation. However, BML requires a shared parametric model across tasks but visual-tactile models for diverse objects have different parameter spaces. To address this issue, we introduce Structured Bayesian Meta-Learning (SBML) that incorporates heterogeneous physics models, enabling learning from training objects with varying appearances and geometries. SBML performs zero-shot vision-only prediction of deformable model parameters and few-shot adaptation after a handful of touches. Experiments show that in two classes of heterogeneous objects, namely plants and shoes, SBML outperforms existing approaches in force and torque prediction accuracy in zero- and few-shot settings. Website: https://shaoxiongyao.github.io/SBML

**Keywords:** Multimodal perception, tactile sensing, few-shot learning

## 1 Introduction

Fusing tactile information with vision is necessary for many manipulation tasks, including reconstructing occluded objects [1], reaching in clutter [2], and assistive dressing [3]. A robot is expected to quickly acquire a visual-tactile model for efficient manipulation. Visual-tactile models of deformable objects such as finite-element models can make accurate predictions when an object's material parameters are properly identified [4, 5, 6]. However, this estimation problem is challenging and usually requires a lot of data. Vision data has significant occlusions, leading to ambiguity in the inference of physical response. Tactile data is noisy and relatively sparse across an object. These characteristics make the identification problems computationally expensive, susceptible to local minima, and ill-posed unless strong assumptions (e.g. homogeneity) are made [7].

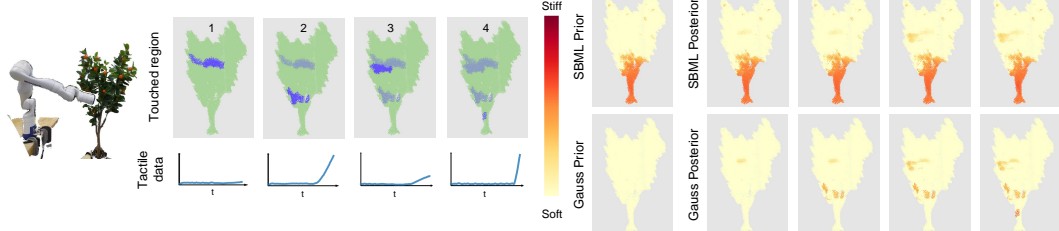

Figure 1: Our SBML approach enables few-shot learning of a novel plant's force response using experience interacting with other plants. Left: touched region (top) and tactile feedback (second row, norm of joint torques over time). Right: compare SBML against a stiffness estimator using naïve uniform Gaussian prior with the same touches. SBML prior already predicts stiffer trunks and branches compared to leaves and its predictions improve as it touches the plant (right 4 columns).

8th Conference on Robot Learning (CoRL 2024), Munich, Germany.

We present a few-shot learning method to estimate dense, heterogeneous visual-tactile models using vision data and a small number of touches. Our approach is based on Bayesian meta-learning (BML) [8, 9], which first learns a *prior distribution over models*, estimated *offline* from the experience of related objects, and then adapts the model *online* via posterior inference. BML is appealing because it can regularize high-capacity (overparameterized) models, which can help estimate heterogeneous visual-tactile models in the face of data sparsity. BML also learns informed priors that improve few-shot accuracy compared to naïve regularization methods. However, applying standard BML requires a unified model parameterization and cannot leverage the underlying physics of different deformable objects. Visual-tactile models for different objects usually have different numbers of state variables and material parameters. For example, the mesh of a large object has more elements than a small object and therefore more parameters.

To enable the application of BML to our setting, we introduce the novel *Structured Bayesian Meta-Learning* (SBML) approach as illustrated in Fig. 2. Each object generates a structure of variable dimension and we learn a unified probabilistic model that predicts material parameters for each element in the structure. The meta-prior is factorized so that it is learned as a parametric function shared across objects, even when the object is segmented into different numbers of parts. We show that SBML can be trained on diverse object sets with to enable rapid and accurate few-shot estimation on novel heterogeneous objects. In summary, this paper makes the following contributions:

- We propose the SBML framework that enables meta-learning on diverse tasks with different numbers of model parameters.
- We show that SBML enables few-shot learning of dense heterogeneous tactile models on real touch data using vision priors and spatial correlation priors.
- We evaluate the proposed approach on real-world plants and shoes datasets. SBML outperforms an unstructured state-of-the-art meta-learning approach and the vanilla VSF (Fig. 1).

## 2 Related Works

**Visual-tactile model estimation**     Fusing visual and tactile information can improve the reliability of robot manipulation [3, 10, 11], but unifying these modalities is challenging due to visual occlusion and tactile sparsity. The parameters of analytical simulations, such as spring-mass and finite element models [5, 12, 6], can be fit to observational data, but these estimation problems are computationally challenging and susceptible to local minima. Learning-based methods have also been applied, but typically assume homogeneity of the object [13, 14, 15] or restricted contact regions during tool usage [16, 17]. These assumptions reduce the representation power (i.e., capacity) of the visual-tactile models. More recently very high-capacity learning-based models (e.g. transformers) have been used to learn latent visual-tactile representation [18, 19], but these models are very data-hungry and are limited to simulation-only or simulation-based pretraining. Our work enables the use of high-capacity heterogeneous visual-tactile models from real-world data by learning an informed prior over model parameters, leading to data-efficient model estimation. We learn prior conditioned on visual information, which has been shown informative in predicting an object's material properties [20, 21, 22] for fabric [23, 24, 25] and rigid surfaces [26, 27, 28]. Our approach goes beyond current methods by predicting heterogeneous material and updating estimation online.

**Meta-learning and Bayesian Meta-learning**     The goal of meta-learning is to simultaneously learn a model that matches a given dataset and a process to adapt the model quickly to novel datasets [29]. Meta-learning has enabled data-efficient robot learning in locomotion [30], manipulation [31, 32] and underactuated control [33, 34]. Past methods often use gradient descent [35, 36] or a learned adaptation module for online update [37, 38, 39]. Bayesian meta-learning uses hierarchical Bayes to create a unified learning objective for all tasks, enabling optimization with a single optimizer and can quantify uncertainty systematically [40, 8, 41, 42, 43]. Whereas existing Bayesian meta-learning methods assume a shared parametric function across all the tasks, our proposed Structured Bayesian Meta-Learning leverages task-specific structures to allow meta-learning over diverse tasks, such as our visual-tactile modeling setting. Meta-learning on graph-structured neural networks [44, 45] is related to our approach but only addresses node classification problems.

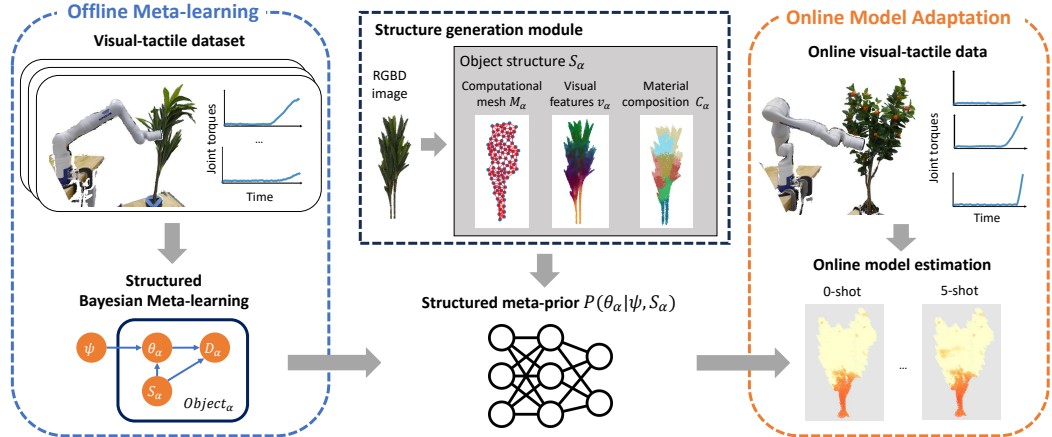

Figure 2: An overview of the structured Bayesian meta-learning method. For each task, a structure generation module uses perception data to create a set of structural assumptions mapping from a common meta-parameter space to visual-tactile models of different objects. Online, maximum a-posteriori estimation is performed to infer material properties over the novel task's structure.

## 3   Method

A visual-tactile model of a deformable object captures how the robot's visual-tactile observations change during an interaction. At time step $t$, the robot executes *action* $a^t$ (e.g., joint position commands) and receives tactile and visual *observations* $z^t$ (e.g. force/torque readings and RGBD images). In our formulation, we consider visual-tactile models that map a sequence of action $a^t$ to a sequence of observations $z^t$ and we denote input-output pair $x = (a^1, \ldots, a^T)$ and $y = (z^1, \ldots, z^T)$ as a *touch*. Visual-tactile model estimation aims to identify this mapping and our SBML approach enables data-efficient estimation with few touches. In this section, we will first present background on BML, our extension to SBML, our instantiation of the framework for visual-tactile modeling, and strategies for making offline and online learning more efficient and accurate.

### 3.1   Structured Bayesian meta-learning

Standard meta-learning considers a meta-training dataset $D = \{D_\alpha \,|\, \alpha = 1, \ldots, L\}$ where $\alpha$ denotes a *task* and $D_\alpha = \{(x_{\alpha,i}, y_{\alpha,i}) \,|\, i = 1, \ldots, N_\alpha\}$ is a task-specific dataset. Each output $y$ follows a distribution $P(y|x, \alpha)$ conditioned on both input $x$ and task $\alpha$, and each $y_{\alpha,i}$ is conditionally independent given $x_{\alpha,i}$ and $\alpha$. The goal of meta-learning is to predict $P(y|x, \alpha_*)$ for a novel task $\alpha_*$ given data from a support set $D_* = \{(x_{*,i}, y_{*,i}) \,|\, i = 1, \ldots, N_*\}$, particularly from the zero-shot ($N_* = 0$) to the few-shot (small $N_*$) cases. Note that $\alpha_*$ is unknown and we need to infer it from the support set $D_*$.

In Bayesian meta-learning the task-conditioned output distribution is approximated as $P(y|x, \alpha) \approx f(y; x, \theta)$, where $\theta \in \mathbb{R}^M$ are the *model parameters* corresponding to the task. A prior $P(\theta)$ is learned offline for online estimation of the model posterior $P(\theta|D_*)$ from the support set. BML considers model parameters of different tasks statistically independent conditioned on meta-parameters $\psi$, i.e. $\theta \sim P(\theta|\psi)$. Note that $f(\cdot; \cdot, \theta)$ must be a sufficiently rich class of functions so that every task maps to a corresponding $\theta$, and $P(\theta|\psi)$ should capture the meta-training data without overfitting. More precisely, offline learning can be formulated as a maximum likelihood estimation of meta-parameters $\psi$ over the meta-training dataset $D$ [8].

$$\hat{\psi} = \arg\max_{\psi} \prod_{\alpha=1}^{L} \int_{\theta_\alpha} P(D_\alpha|\theta_\alpha) P(\theta_\alpha|\psi) d\theta_\alpha. \tag{1}$$

Here the data likelihood is evaluated as $P(D_\alpha|\theta_\alpha) = \prod_{i=1}^{N_\alpha} f(y_{\alpha,i}; x_{\alpha,i}, \theta_\alpha)$. In the online phase, we find the maximum a posteriori (MAP) estimate of model parameters $\hat{\theta}_*$ using the Bayes rule:

$$\hat{\theta}_* = \arg\max_{\theta_*} P(D_*|\theta_*)P(\theta_*|\psi) \tag{2}$$

and then the online prediction is given by $P(y|x, D_*) \approx f(y; x, \hat{\theta}_*)$.

In the visual-tactile model estimation setting, the standard assumption in BML regarding fixed-dimensional inputs $x$, outputs $y$, and model parameters $\theta$ does not hold. Different touches have different sequence lengths $T_{\alpha,i}$ and diverse objects have varying dimensional model parameters $\theta_\alpha \in \mathbb{R}^{M_\alpha}$. In particular, we represent a heterogeneous deformable object as a computational mesh with various particles and elements, leading to varying numbers of material parameters to estimate.

Our proposed Structured Bayesian Meta-Learning (SBML) framework addresses this problem using task-specific Bayesian network structures mapping a fixed-dimensional meta-parameter to a prior that supports diverse tasks. The structures for each task $\alpha$ are generated by a *structure generation module* outside the learning pipeline as in Fig. 2. We use $\mathcal{S}_\alpha$ to denote the task-dependent structure. Conditioned on $\mathcal{S}_\alpha$, the probability distributions in Eq. (1) and (2) are factorized into parametric functions of $\psi$ shared across tasks. We consider two types of structures specifically relevant to visual-tactile models. First, a *hidden Markov structure* can handle the dynamics and variation in sequence length $T_{\alpha,i}$ governing the dimensionality of $x_{\alpha,i} = (a_{\alpha,i}^1, \ldots, a_{\alpha,i}^{T_{\alpha,i}})$ and $y_{\alpha,i} = (z_{\alpha,i}^1, \ldots, z_{\alpha,i}^{T_{\alpha,i}})$. In this model, the object state $s^t$ has a Markovian transition model defined by a dynamics model $Dyn$ and the observations $z^t$ depend on the current state $s^t$ via an observation model $P(z^t|s^t, \theta_\alpha, \mathcal{S}_\alpha)$. The data likelihood $f(y_{\alpha,i}; x_{\alpha,i}, \theta_\alpha, \mathcal{S}_\alpha)$ is therefore factorized as follows:

$$f(y_{\alpha,i}; x_{\alpha,i}, \theta_\alpha, \mathcal{S}_\alpha) = \prod_{t=1}^{T_{\alpha,i}} P(z_{\alpha,i}^t|s_{\alpha,i}^t, \theta_\alpha, \mathcal{S}_\alpha), \quad \text{where } s_{\alpha,i}^t = Dyn(s_{\alpha,i}^{t-1}, a_{\alpha,i}^t; \theta_\alpha, \mathcal{S}_\alpha). \tag{3}$$

Here $\mathcal{S}_\alpha$ provides the initial state $s_{\alpha,i}^0$. For simplicity, we consider a dynamic model $Dyn$ that is deterministic given material parameters $\theta_\alpha$. From this formulation we can replace $f$ in Eq. (1) and (2) with its structured counterpart.

Second, SBML addresses the structure-dependent set of model parameters $\theta_\alpha$ and the prior in Eq. (1) and (2) changes from $P(\theta_\alpha|\psi)$ to $P(\theta_\alpha|\psi, \mathcal{S}_\alpha)$. We formulate a *decoupled latent variable structure* that maps a set of latent parameters $\phi_\alpha$ to a task-dependent prior over model parameters $\theta_\alpha$, via a linear map $\theta_\alpha = B_\alpha \phi_\alpha$ for simplicity. The structure $\mathcal{S}_\alpha$ defines a set of features $v_\alpha$ informative to $\phi_\alpha$, which are visual features $v_{\alpha,j}$ mapped to each element $j$ for a visual-tactile model. The latent parameters $\phi_\alpha$ has a factorized distribution conditioned on $v_\alpha$, i.e. $\phi_\alpha|v_\alpha \sim \prod_{k=1}^{|\phi_\alpha|} P(\phi_{\alpha,k}|v_{\alpha,k}, \psi)$. Ultimately, SBML learns parametric functions $P(\phi_{\alpha,k}|v_{\alpha,k}, \psi) = g(\phi_{\alpha,k}; v_{\alpha,k}, \psi)$ shared between tasks, e.g. a neural network that outputs a probability density. The latent structure depends on the instantiation of this framework, and our implementation is described below.

### 3.2 Visual-tactile model estimation via SBML

In a visual-tactile model, we consider structure $\mathcal{S}_\alpha$ that captures an object's geometry in a computational mesh $\mathcal{M}_\alpha$, with initial state $s_\alpha^0$, its appearance in visual features $v_\alpha$ and material correlation in composition structure $\mathcal{C}_\alpha$ as in Fig. 3. We will define the visual-tactile simulation model and material parameters $\theta_\alpha$ to evaluate data likelihood in Eq. (3). The material parameter prior maps the universal latent parameters prior $g(\phi_{\alpha,k}; v_{\alpha,k}, \psi)$ to $P(\theta_\alpha|\psi, \mathcal{S}_\alpha)$ in a manner that defines a relationship between material parameters and appearance, and correlations between elements.

**Object representation and simulation model.** We consider particle-based visual-tactile models that discretize the object $\alpha$ into $n_\alpha$ particles. The *state* of the object is the positions of the particles $s^t \in \mathbb{R}^{3n_\alpha} = (p_1^t, ..., p_{n_\alpha}^t)$, where $p_k^t \in \mathbb{R}^3$ is particle $k$'s position at time step $t$. The interaction between particles is defined by a computational mesh $\mathcal{M}_\alpha$ containing $m_\alpha$ elements, and each element specifies a subset of particles with mutual interactions. The $j$th element, $j = 1, \ldots, m_\alpha$, has a fixed $d$-dimensional material parameter $\theta_{\alpha,j} \in \mathbb{R}^d$ that enables us to simulate particle interactions and observations. The dynamics model $Dyn$ updates the particle deformation $s^t$ using interactions in each element as in Fig. 3. The visual-tactile model parameters of the object are the tuple of material parameters of each element $\theta_\alpha = (\theta_{\alpha,1}, ..., \theta_{\alpha,m_\alpha})$.

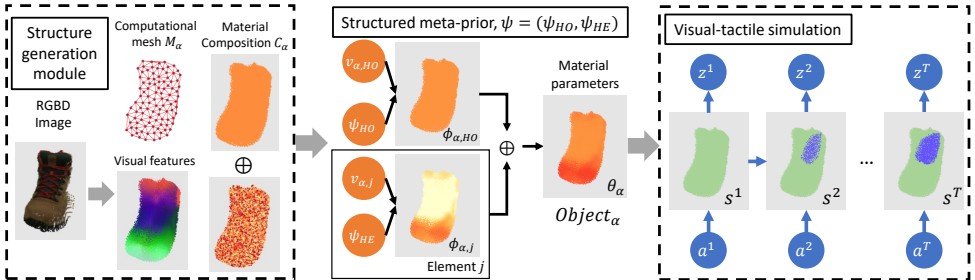

Figure 3: Overview of structures in visual-tactile model estimation. Left: construct the structure $S_\alpha$ for an object; Middle: map meta-parameters $\psi$ to material parameters $\theta_\alpha$ with homogeneous + heterogeneous composition structure; Right: temporal structure in visual-tactile simulation.

**Visual features and composition structure.** We use visual features and composition structures to define the map from latent parameters prior to primary material prior. This allows our model to capture the relationship between appearance and material, and the correlation between material parameters.

First, $v_\alpha$ defines visual features mapped to each element in $\mathcal{M}_\alpha$, where element $j$ has feature $v_{\alpha,j}$. Choosing expressive visual features will make the prior more informative to predict material parameters, we use dense features from pre-trained vision models DINOv2 [46] by default.

Second, the material composition structure $\mathcal{C}_\alpha$ lets us accommodate correlation assumptions. The composition structure assumes the object is composed of components with similar materials, e.g. branches on a tree share similar high stiffness compared to soft leaf regions. We define a set of components and each component has material parameters $\phi$ mapped to a subset of elements $c \subset \mathcal{M}_\alpha$. Here component material parameters $\phi$ are the decoupled latent variables. We consider three types of components denoted by *component type* $b \in \{HE, HO, SEG\}$, where *HEterogeneous* component has a single element, *HOmogeneous* component has all elements, and *SEGmented* component has a cluster of elements. Also, the visual features for a single component aggregate the visual features $v_{\alpha,j}$ of all elements $j \in c$ and we use the average feature $\frac{1}{|c|} \sum_{j \in c} v_{\alpha,j}$ for simplicity.

Formally, $\mathcal{C}_\alpha = \{(c_{\alpha,k}, b_{\alpha,k}, \phi_{\alpha,k}, v_{\alpha,k})\}_{k=1,\ldots,|\mathcal{C}_\alpha|}$ defines a set of components by their elements, types, component material parameters, and visual features. Then, the material parameters are the superposition of components $\theta_{\alpha,j} = \sum_{k=1}^{|\mathcal{C}_\alpha|} \mathbb{I}[j \in c_{\alpha,k}]\phi_{\alpha,k}$ and we define $\theta_\alpha = B_\alpha \phi_\alpha$, where $B_\alpha$ is a $(0,1)$ matrix of size $M_\alpha \times |\mathcal{C}_\alpha|$ and $\phi_\alpha \equiv (\phi_{\alpha,1}, \ldots, \phi_{\alpha,|\mathcal{C}_\alpha|})$ stacks all component material parameters. Our prior expresses the knowledge that components of the same type share the same material parameters distribution conditioned on the visual feature, $P(\phi_{\alpha,k}|v_{\alpha,k}, \psi) = g_{b_{\alpha,k}}(\phi_{\alpha,k}; v_{\alpha,k}, \psi)$. We provide different types of components with independent priors and with an independent subset of meta-parameters: $g_{b_{\alpha,k}}(\phi_{\alpha,k}; v_{\alpha,k}, \psi) \equiv g_{b_{\alpha,k}}(\phi_{\alpha,k}; v_{\alpha,k}, \psi_{b_{\alpha,k}})$. The material parameters prior can be evaluated by marginalizing over component material parameters $\phi_\alpha$,

$$P(\theta_\alpha|\psi, \mathcal{S}_\alpha) = \int_{\phi_\alpha} \mathbb{I}[\theta_\alpha = B_\alpha \phi_\alpha] \prod_{k=1}^{|\mathcal{C}_\alpha|} g_{b_{\alpha,k}}(\phi_{\alpha,k}; v_{\alpha,k}, \psi)d\phi_\alpha. \tag{4}$$

Overall, Eqs. (3) and (4) enable SBML to define a universal meta-learning problem over $\psi$ and task-dependent posterior estimation over $\theta_\alpha$ regardless of the tasks' structure.

### 3.3 Efficient implementation with Gaussian likelihoods

To achieve efficient implementation, we instantiate the SBML framework on the Volumetric Stiffness Field(VSF) [47] visual-tactile model. VSF is a dense particle-based representation with independent Hookean springs and can flexibly represent heterogeneous material.

- In Eq. (3), the dynamics model $Dyn$ is a particle-based deformation simulator the same as [47] and the observation model $P(z^t|s^t, \theta_\alpha, \mathcal{S}_\alpha)$ is approximated as a Linear-Gaussian function. Linearity is from the Hookean springs in VSF and the additivity of forces/torques over springs. The Gaussian assumption models the tactile sensor noise as a normal distribution and is a reasonable approximation for the sensors we evaluated.

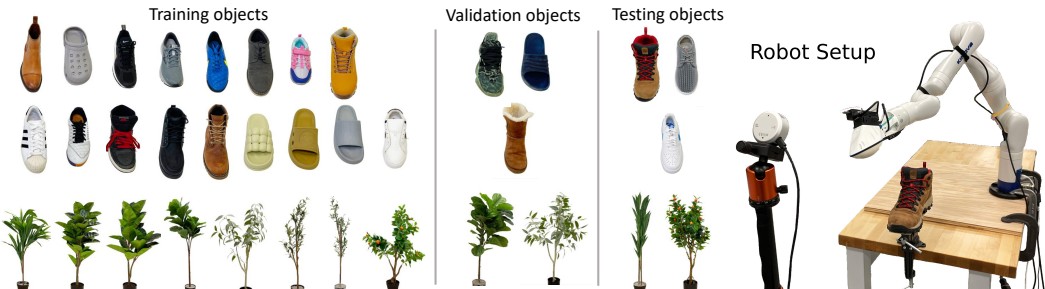

Figure 4: Left: Objects in the *shoes* (top) and *plants* (bottom) datasets. The scales of the shoes and plants are different; Right: Physical experiment setup collecting *shoes* dataset with Punyo tactile sensor and Kinova Gen3 arm.

- The prior $P(\theta_\alpha|\psi, \mathcal{S}_\alpha)$ defines a Gaussian distribution over the material parameter $\theta_\alpha$. In particular, $g_{b_{\alpha,k}}$ is a neural network that outputs the mean and variance of $\phi_{\alpha,k}$. The integration in Eq. (4) has a closed-form solution with a Gaussian integrand.

We evaluate likelihood in Eq. (1) using PyTorch [48] and optimize meta-parameters $\psi$ via gradient descent. The online parameter estimation in Eq. (2) is a quadratic program (QP) with nonnegativity constraints on the spring stiffnesses. Further details are provided in Appendix A.

## 4 Experiments and Results

We acquire two benchmark datasets using the experimental platform in Fig. 4: 1) *Plants* dataset has 12 artificial plants each touched 50-70 times and 7-d joint torque from the Kinova arm; 2) *Shoes* dataset has 23 shoes each touched 20-30 times and pressure data from a Punyo tactile sensor [49]. We use joint torques because the robot arm contacts the plants along its entire arm, while contact with shoes is localized to the end-effector. We show that SBML enables data-efficient visual-tactile model estimation by comparing SBML's zero- and few-shot predictions with baselines on real-world datasets. We also evaluate the effects of different meta-training datasets and material composition structures on SBML performance. Further experiments on the choice of visual features, prior capacity, and dataset size are presented in Appendix C.

### 4.1 SBML implementation on VSF model

**Structure generation module**  The structure generation module samples VSF particles from the object's surface and interior using ray casting. Visual features $v_\alpha$ are generated by DINOv2 [46] and projected in 3D space. Our standard setting assumes a heterogeneous material composition $\mathcal{C}_\alpha$. Other material composition structures are tested in Sec. 4.5.

**Meta-prior**  The meta-prior for heterogeneous components uses a multi-layer perceptron (MLP) that outputs the mean and standard deviation of a Gaussian distribution. The meta-priors for the segment and homogeneous components in Sec. 4.5 use a linear mean model with learnable variance, chosen due to the small number of segments and tasks in the meta-training dataset.

**Observation models**  The observation $z^t$ is joint torques $\tau^t \in \mathbb{R}^7$ in *plants* and pressure difference from no contact $\delta^t \in \mathbb{R}$ in *shoes*. Note the observation models are linear-Gaussian in the spring stiffness. Joint torques are the summation of Jacobian transpose times contact forces. For the Punyo sensor, we use a simplified 1-D observation model that sums the force magnitude of each point.

### 4.2 Baseline methods

**Non-structured meta-learning**  We compare a standard meta-learning method that directly maps robot action to tactile observation using no object-specific structure. The model is instantiated as MLP and uses DINOv2 classification token [46] to encode visual information. The model is meta-trained and adapted using iMAML [36], where the inner loop adaptation uses gradient descent.

**Structured model with naïve prior**  We compare to the "vanilla" VSF model with a naïve Gaussian prior. Vision is not used, and the naïve prior assigns the same mean and variance to all particles. The mean and variance are estimated from estimated VSF stiffness in the meta-training dataset.

## 4.3 Zero-shot stiffness estimation

We visualize the zero-shot stiffness prediction of SBML for test objects in Fig. 5 organized in three rows: the top row shows the object, the middle row shows the SBML prior mean and the bottom row shows the estimated stiffness using vanilla VSF (average prior) given all touches in the dataset. We observe that the zero-shot stiffness estimates are qualitatively quite close to the all-touch estimates, with stiffer estimates at the branches and trunk of plants and the toes of shoes. Note that VSF all-shot is an estimation with all available support data and we cannot get the precise material parameters. The uncertainty of SBML prior is visualized in Fig. 10.

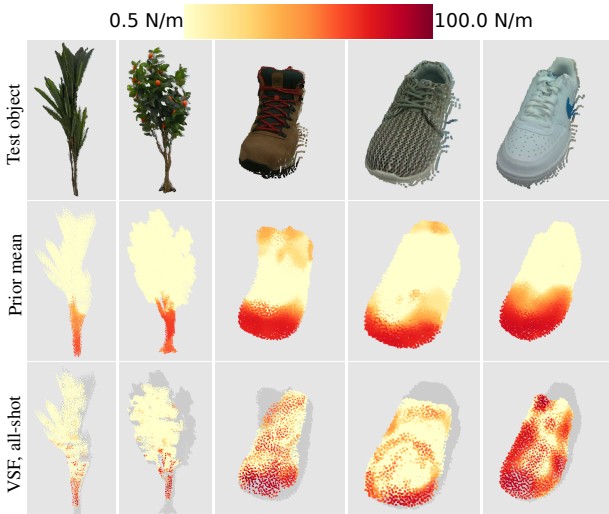

Figure 5: Learned zero-shot priors vs estimated VSF from all touches (w/o prior). Points never touched in the dataset are shown in gray. Stiffness color map is in the log scale.

## 4.4 Breadth of meta-training set

Next, we show the SBML $k$-shot predictions and evaluate how the breadth of the meta-training set affects results. We expect the best performance when the meta-training set distribution is narrow and contains the test object within its support. The broad datasets contain all training objects as shown in Fig. 4. The *plants* narrow meta-training dataset contains different views of the same plant and the test set contains touches on the novel view. For the *shoes* dataset, we create narrow datasets by selecting shoes of similar materials, specifically choosing two boots, two nylon running shoes, and two sneakers for each test object.

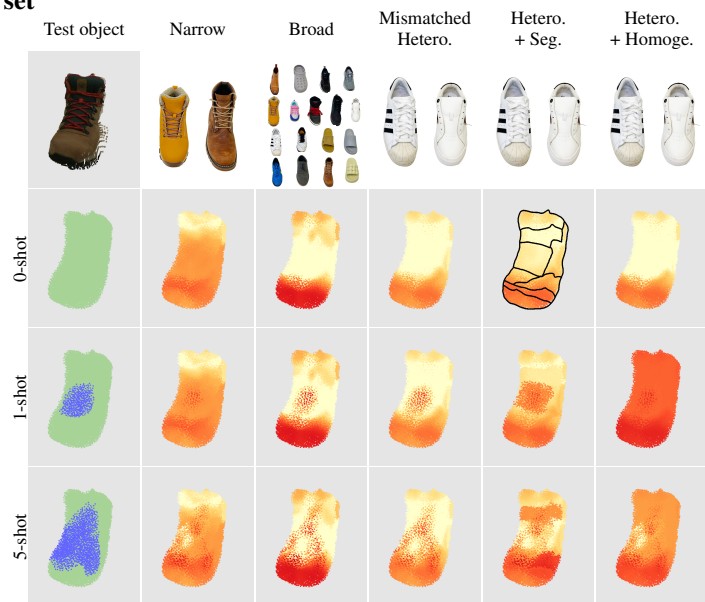

Figure 6: Effects of the meta-training dataset on $k$-shot predictions. The first column shows the touched region in blue and the untouched region in green. Same colormap to Fig. 5. Black lines in (Hetero.+Seg. column, 0-shot row) show segment boundaries. See Sec. 4.5 for details.

Fig. 6 shows the qualitative results for the boot test object, where SBML is trained with *narrow*, *broad*, and *mismatched* meta-training sets. The $k$-shot row indicates the posterior estimate after observing a support set of $k$ touching sequences from the test object's dataset. In the second row of Fig. 6, all priors estimate the front of the toe to be stiff, but only the narrow prior includes a stiff tongue. The broad prior has a moderately low stiffness and the mismatched prior estimates a very low stiffness due to the large domain gap. As expected, each model adapts to the observed data, but the narrowly trained model tends to shift less.

We show quantitative results on the test objects in Tab. 1 and Tab. 2. The prediction accuracy is evaluated on a *query set* disjoint from the support set. We also report the standard deviation of

Table 1: Joint torque prediction error (Nm) ± std over query set on Plant benchmark.

| | iMAML (broad dataset) | Gaussian prior (average) | SBML prior (narrow dataset) | SBML prior (broad dataset) |
|---|---|---|---|---|
| 0-shot | $3.86 \pm 0.84$ | $5.23 \pm 0.00$ | $2.82 \pm 0.04$ | $\mathbf{2.80 \pm 0.05}$ |
| 1-shot | $4.99 \pm 2.04$ | $5.04 \pm 0.04$ | $2.78 \pm 0.03$ | $\mathbf{2.76 \pm 0.05}$ |
| 5-shot | $4.63 \pm 1.26$ | $4.42 \pm 0.07$ | $2.71 \pm 0.05$ | $\mathbf{2.67 \pm 0.05}$ |
| 10-shot | $4.51 \pm 1.03$ | $3.88 \pm 0.09$ | $2.66 \pm 0.03$ | $\mathbf{2.61 \pm 0.04}$ |
| VSF all-shot 2.57 | | | | |

Table 2: Tactile prediction error (hPa) ± std over query set on Shoe benchmark.

| | iMAML (broad dataset) | Gaussian prior (average) | SBML prior (narrow dataset) | SBML prior (broad dataset) |
|---|---|---|---|---|
| 0-shot | $9.95 \pm 6.71$ | $8.27 \pm 0.00$ | $\mathbf{5.66 \pm 0.23}$ | $7.43 \pm 0.25$ |
| 1-shot | $6.30 \pm 1.00$ | $7.46 \pm 0.13$ | $\mathbf{5.48 \pm 0.19}$ | $6.77 \pm 0.26$ |
| 5-shot | $6.21 \pm 0.96$ | $5.54 \pm 0.21$ | $\mathbf{4.91 \pm 0.12}$ | $5.05 \pm 0.20$ |
| 10-shot | $6.15 \pm 0.91$ | $4.62 \pm 0.08$ | $4.54 \pm 0.07$ | $\mathbf{4.18 \pm 0.04}$ |
| VSF all-shot 4.10 | | | | |

prediction error over 10 random seeds. The Gaussian prior has zero standard deviation for 0-shot since its initial mean prediction is deterministic. The last row shows the VSF prediction error using all support set touches covering the entire object.

SBML not only demonstrates lower zero-shot prediction errors in the first row of Tab. 1 and Tab. 2, underscoring the value of incorporating visual information but also quickly improves with additional support data, achieving near-all-shot accuracy by 10 shots. SBML's absolute improvement is smaller due to an almost optimal zero-shot error in Tab. 1. In contrast, iMAML exhibits worse zero-shot performance and noisy adaptation. Using object-specific structures, the Gaussian prior improves the estimation consistently but has worse few-shot accuracy due to naive uniform mean. Comparing meta-training datasets, we observe that SBML performs best when the training and testing distributions closely align. While the performance gap across meta-training distributions is minor on *plants*, it becomes significantly larger on *shoes*, likely due to the greater variation in shoes' appearance and material. Shoes are made of brown leather in boots, gray nylon in sneakers, and various colored plastics in slippers, while plants typically have brown branches and green leaves, as shown in Fig. 4. To evaluate the capacity of our method, we trained a unified prior using data from both plant and shoe datasets. The results in Appendix C.3 demonstrate that our method has sufficient capacity to learn a prior shared between the plant and shoe categories.

## 4.5 Effects of material composition structure

Finally, we evaluate the effects of changing the material composition structure and we tested heterogeneous, heterogeneous + segment, and heterogeneous + homogeneous. Qualitative results are shown in the right three columns of Fig. 6. We can see that the stiffness across a segment changes together for heterogeneous + segment, and the stiffness of the entire object changes for heterogeneous + homogeneous. We also found the composition structure speeds up model adaptation quantitatively in Tab. 3. When the test object is out of meta-training distribution, the learned stiffness prior will predict inaccurate stiffness values, and the shared components will help learn across regions of coherent material. We test these composition structures on narrow and broad datasets, but the improvement is marginal.

| | Hetero. | Hetero. + Seg. | Hetero. + Homoge. |
|---|---|---|---|
| 0-shot | $9.76 \pm 0.75$ | $9.23 \pm 0.86$ | $\mathbf{9.13 \pm 0.77}$ |
| 1-shot | $8.58 \pm 0.68$ | $\mathbf{7.57 \pm 0.57}$ | $8.10 \pm 1.38$ |
| 5-shot | $5.82 \pm 0.31$ | $\mathbf{5.00 \pm 0.26}$ | $5.32 \pm 0.17$ |
| 10-shot | $4.73 \pm 0.10$ | $\mathbf{4.36 \pm 0.07}$ | $4.49 \pm 0.07$ |
| VSF all-shot 4.10 | | | |

Table 3: SBML's predictions with different compositions on shoes *mismatched* dataset.

## 5 Conclusion and Limitations

We present a novel structured Bayesian meta-learning (SBML) approach to address the few-shot learning of visual-tactile models. Our innovations allow the method to transfer offline knowledge from objects of different sizes and shapes to improve prediction accuracy on novel objects. The method is highly flexible and can accommodate different material compositions, feature representations, and model capacities. Applied to benchmark datasets of plants and shoes, our experiments demonstrate that it outperforms non-informed estimators as well as non-structured meta-learning approaches in zero- and few-shot prediction accuracy.

**Limitations:** The SBML framework is evaluated on the VSF model that can only predict tactile response, and we are interested in extending it to other deformable object models such as FEM and Graph Neural Dynamics Models that can predict visual deformation. The Gaussian prior assumes a single mode of material parameters conditioned on visual features and cannot handle more complicated multi-modal distributions. Our work has only demonstrated generalization within object categories, but this approach has not yet enabled generalization between categories.

**Acknowledgments**

This work has partially been funded by USDA/NIFA Grant #2020-67021-32799, and we thank Toyota Research Institute for a loan of the Punyo tactile sensor. We thank Joao M. C. Marques, Jing-Chen Peng, Mengchao Zhang, Patrick Naughton, and Sicong Pan for their valuable comments and suggestions. We are grateful to Yidi Hua for segmenting object images. We also thank Simon Kato, Haonan Chen, Yuan Shen, Zhen Zhu, Shuijing Liu, Hameed Abdul-Rashid, Yiqiu Sun, Mengchao Zhang, Yangfei Dai, Yixuan Wang, Kaiwen Hong, Shuhong Zheng, and Yuxiang Liu for providing objects used in the experiments.

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

# Appendices

## A  Efficient visual-tactile model estimation

Here we provide more details on the efficient SBML implementation on VSF with Gaussian likelihoods, outlined in Sec. 3.3.

**Volumetric stiffness field**  A VSF defines a volume of independent particles that resist displacements from their rest position with Hookean springs. (Note that this differs from standard spring-mass systems where springs connect particles). The dynamics model $Dyn$ is a geometric simulator that simulates the particle displacements independently. This simulator detects collisions between the robot surface and particles and simulates the stick-slip motion of particles. Despite its simplicity, since the particle's displacement is independent, this deformation model enables highly efficient estimation by allowing parallel estimation of stiffness values.

The structure defined by a VSF consists of a computational mesh $\mathcal{M}_\alpha$ with $n_\alpha$ particles and $m_\alpha = n_\alpha$ springs. Each particle responds to displacement with a Hookean reactive force $f_i^t = -K_{\alpha,i} \cdot (p_i^t - p_i^0)$, with $K_{\alpha,i}$ the stiffness of this spring. In this case, the material parameters are the stiffness of each spring $\theta_\alpha = (K_{\alpha,1}, ..., K_{\alpha,n})$.

Most consequentially, the VSF implementation leads to a material-independent dynamics model $Dyn$ in Eq. 3:

$$s_{\alpha,i}^t = Dyn(s_{\alpha,i}^{t-1}, a_{\alpha,i}^t; \mathcal{S}_\alpha). \tag{5}$$

where $s_{\alpha,i}^t \in \mathbb{R}^{3n_\alpha}$ represents the particle positions and we consider particles at rest positions for $s_{\alpha,i}^0$. Material-independence is a unique characteristic of the VSF model and the material parameters influence the data likelihood only through the observation model $P(z^t|s^t, \theta_\alpha, \mathcal{S}_\alpha)$.

**Linear-Gaussian observation model**  Next, we approximate the observation likelihood as a linear-Gaussian model, linearly dependent on the material parameters. Here the linearity is from Hookean spring in VSF. We assume $z^t = W^t \theta_\alpha + \epsilon_z^t$ with $\epsilon_z^t \sim N(\mu^t, \Sigma_z)$ a Gaussian noise term with known covariance. Without loss of generality, we will assume $\mu^t = 0$ to simplify the subsequent notation. Here $W^t$ is a known observation matrix that is a nonlinear function of state $s^t$, i.e. $W^t = W(s^t)$. In other words,

$$P(z^t|s^t, \theta_\alpha, \mathcal{S}_\alpha) \approx N(z^t; W^t \theta_\alpha, \Sigma_z). \tag{6}$$

For the plant dataset, the linear model of joint torques observation $z^t = \tau^t$ can be evaluated as,

$$\tau^t = \sum_i -J(p_i^t, q^t)^\top K_{\alpha,i} \cdot (p_i^t - p_i^0) + \epsilon_\tau, \ W^t = \begin{bmatrix} -J_1^\top (p_1^t - p_1^0) & \cdots & -J_{n_\alpha}^\top (p_{n_\alpha}^t - p_{n_\alpha}^0) \end{bmatrix}. \tag{7}$$

Here $W^t$ has dimension $7 \times n_\alpha$ and Jacobian matrix $J(p_i^t, q^t) = J_i$ is computed from forward kinematics at robot configuration $q^t$ and $\epsilon_\tau$ is Gaussian noise in joint torques. For the shoe dataset, we used a simplified 1-D force model to model the pressure difference $z^t = \delta^t$:

$$\delta^t = \sum_i K_i \cdot \|p_i^t - p_i^0\|_2 + \epsilon_p, \ W^t = \begin{bmatrix} \|p_1^t - p_1^0\|_2 & \cdots & \|p_{n_\alpha}^t - p_{n_\alpha}^0\|_2 \end{bmatrix}. \tag{8}$$

Here $W^t$ has dimension $1 \times n_\alpha$ and $\epsilon_p$ is Gaussian noise in pressure difference. Now, the log-likelihood of a sequence of observations defined in Eq. (3) has a quadratic form in $\theta_\alpha$:

$$-\log f(y_{\alpha,i}; x_{\alpha,i}, \theta_\alpha) = -\log \prod_{t=1}^{T_{\alpha,i}} P(z_{\alpha,i}^t|s_{\alpha,i}^t, \theta_\alpha) = \frac{1}{2} \sum_{t=1}^{T_{\alpha,i}} \|z_{\alpha,i}^t - W_{\alpha,i}^t \theta_\alpha\|_{\Sigma_z^{-1}}^2 + const. \tag{9}$$

**Gaussian material parameters prior**  We assume the component material parameters prior $g_{b_{\alpha,k}}(\phi_{\alpha,k}; v_{\alpha,k}, \psi)$ defines a Gaussian distribution over $\phi_{\alpha,k}$. We use a probabilistic neural network that takes the visual feature $v_{\alpha,k}$ as input and outputs the mean $\mu_{\alpha,k}$ and variance $\Sigma_{\alpha,k}$, where $\psi$ are the weights in the network.

$$g_{b_{\alpha,k}}(\phi_{\alpha,k}; v_{\alpha,k}, \psi) = N(\phi_{\alpha,k}; \mu_{\alpha,k}, \Sigma_{\alpha,k}). \tag{10}$$

With this definition, the mean and variance of each element's parameter are sums of Gaussians, which are themselves Gaussian. Let $\mu_\alpha = (\mu_{\alpha,1}, ..., \mu_{\alpha,|\mathcal{C}_\alpha|})$ and $\Sigma_\alpha = \text{diag}(\Sigma_{\alpha,1}, ..., \Sigma_{\alpha,|\mathcal{C}_\alpha|})$, $\phi_\alpha$ follows a Gaussian distribution with mean $\mu_\alpha$ and covariance $\Sigma_\alpha$:

$$\phi_\alpha \sim N(\phi_\alpha; \mu_\alpha, \Sigma_\alpha). \tag{11}$$

We denote linear map from $\phi_\alpha$ to $\theta_\alpha$ as $\theta_\alpha = B_\alpha \phi_\alpha$ in Sec. 3.1, where $B_\alpha[j,k] = \mathbb{I}[j \in c_{\alpha,k}]$. Because $\theta_\alpha$ is a linear transformation of Gaussian random variable $\phi_\alpha$, the material parameters prior in Eq. (4) has a closed-form evaluation in this case [50]:

$$P(\theta_\alpha | \psi, \mathcal{S}_\alpha) = \int_{\phi_\alpha} \mathbb{I}[\theta_\alpha = B_\alpha \phi_\alpha] \prod_{k=1}^{|\mathcal{C}_\alpha|} g_{b_{\alpha,k}}(\phi_{\alpha,k}; v_{\alpha,k}, \psi) d\phi_\alpha$$

$$= \int_{\phi_\alpha} \mathbb{I}[\theta_\alpha = B_\alpha \phi_\alpha] N(\phi_\alpha; \mu_\alpha, \Sigma_\alpha) d\phi_\alpha = N(\theta_\alpha; B_\alpha \mu_\alpha, B_\alpha \Sigma_\alpha B_\alpha^\top) \tag{12}$$

**Offline MLE via SGD**  Evaluating structured Bayesian meta-learning MLE objective in Eq. (1) requires integration over material parameters $\theta_\alpha$. From the deterministic dynamics model $Dyn$ in Eq. (5), we can get state trajectory $(s_{\alpha,i}^0, ..., s_{\alpha,i}^{T_{\alpha,i}})$ from action sequence. The observation matrix can be evaluated as $W_{\alpha,i}^t = W(s_{\alpha,i}^t)$ using Eq. (7) or (8). The task dataset likelihood in Eq. (1) can be evaluated in closed-form:

$$P(D_\alpha | \theta_\alpha) = \prod_{i=1}^{N_\alpha} f(y_{\alpha,i}; x_{\alpha,i}, \theta_\alpha) = \prod_{i=1}^{N_\alpha} \prod_{t=1}^{T_{\alpha,i}} N(z_{\alpha,i}^t; W_{\alpha,i}^t \theta_\alpha, \Sigma_z) = N(z_\alpha; W_\alpha \theta_\alpha, \Sigma_Z). \tag{13}$$

Here we stack all observations in $z_\alpha = (z_{\alpha,1}^1, ..., z_{\alpha,N_\alpha}^{T_{\alpha,N_\alpha}})$ into a single vector and all observation matrices $W_\alpha = (W_{\alpha,1}^1, ..., W_{\alpha,N_\alpha}^{T_{\alpha,N_\alpha}})$ into a single matrix. The observation noise matrix $\Sigma_Z = \text{diag}(\Sigma_z, ..., \Sigma_z)$ is block diagonal matrix with each observation noise matrix. The number of observations $\sum_{i=1}^{N_\alpha} T_{\alpha,i}$ is on the order of $10^4$ for each task and $(z_\alpha, W_\alpha)$ can be loaded on GPU. The log-likelihood function in Eq. (1) on the entire dataset can be evaluated as,

$$\sum_{\alpha=1}^{L} \log \int_{\theta_\alpha} P(D_\alpha | \theta_\alpha) P(\theta_\alpha | \mathcal{S}_\alpha, \psi) d\theta_\alpha$$

$$= \sum_{\alpha=1}^{L} \log \int_{\theta_\alpha} N(z_\alpha; W_\alpha \theta_\alpha, \Sigma_Z) N(\theta_\alpha; B_\alpha \mu_\alpha, B_\alpha \Sigma_\alpha B_\alpha^\top) d\theta_\alpha \tag{14}$$

Using the fact that the marginalization over a Gaussian distribution is still Gaussian, the offline MLE problem in Eq. (1) becomes

$$\hat{\psi} = \arg\min_\psi \sum_{\alpha=1}^{L} \frac{1}{2} \|z_\alpha - W_\alpha B_\alpha \mu_\alpha\|_{(W_\alpha B_\alpha \Sigma_\alpha B_\alpha^\top W_\alpha^\top + \Sigma_Z)^{-1}}^2$$

$$+ \frac{1}{2} \log \det \left(W_\alpha B_\alpha \Sigma_\alpha B_\alpha^\top W_\alpha^\top + \Sigma_Z\right) \tag{15}$$

Here we minimize the negative log-likelihood and omit constant terms. This objective function is implemented using differentiable operations in PyTorch [48] and the optimization is executed using stochastic gradient descent. We use the Adam [51] optimizer with learning rate $10^{-4}$ to optimize the meta-parameters.

**Online MAP as QP**  Because of Gaussian prior and linear-Gaussian observation model, we instantiate the online MAP problem in Eq. (2) as a QP:

$$\hat{\theta}_* = \arg\min_{\theta_*} \|z_* - W_* \theta_*\|_{\Sigma_Z^{-1}}^2 + \|\theta_* - B_* \mu_*\|_{(B_* \Sigma_* B_*^\top)^{-1}}^2, \quad \text{s.t. } \theta_* \geq 0 \tag{16}$$

Here $z_*$ and $W_*$ are evaluated on the support set $D_*$. The non-negative constraint is from the non-negative spring stiffness $\theta_{\alpha,i} = K_{\alpha,i} \geq 0$. Finally, the estimated material parameters $\hat{\theta}_*$ can be used to predict tactile response for a novel touch. In our implementation, the quadratic program in Eq. (16) is solved online using CVXPY [52].

**Avoiding meta-overfitting**   Visual-tactile datasets are relatively small, and meta-learning is prone to meta-overfitting where the meta-parameters memorize all the training tasks, so it is important to adopt measures to avoid overfitting. We use several strategies in our work:

- Early stopping when the validation error increases.
- Adding dropout layers in the meta-prior network.
- A good prior initialization was important to avoid over-confident priors. We initialize the meta-prior uncertainty using the estimated stiffness standard deviation from the training dataset.

Although regularization via a meta-prior-prior has been proposed for Bayesian meta-learning [53], we find that it is challenging to define appropriate regularization targets for the prior weights that discourage overconfidence. Detailed training configurations are presented in the Appendix B.

## B   Experimental setup

### B.1   Equipment and benchmark datasets

The objects are fixed to a table within the robot's workspace and the robot touches the object multiple times. A static RGBD camera is used to capture a view of each object before each touch. A time series of touch data is recorded during each touch.

The *Plants* dataset consists of 12 artificial plants with different sizes, appearances, and materials. A Kinect Azure is used to capture RGBD data. We collect the 7-d joint torque data from the Kinova arm as the touch data. For each plant, we rotate the object in 4 different views (Fig. 7) and sample 50–70 touches for each view. There are a total of $3,255$ touching sequences and $807,103$ individual touch readings.

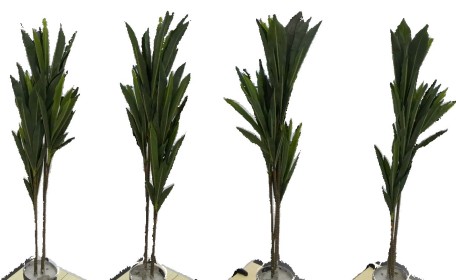

Figure 7: Four different views of the artificial dracaena in *plant* dataset.

The *Shoes* dataset consists of 23 shoes including boots, running shoes, sneakers, and slippers. An Intel Realsense L515 is used to capture RGB-D data. Pressure data is collected with a Punyo tactile sensor [49]. 23 shoes have very different materials and visual appearances. We sample the touched point uniformly at random and select the touch direction along the surface normal. We touch each shoe 20–30 times. There are a total of 729 touching sequences and $35,614$ individual touch readings.

For each object in the dataset, we sample target points uniformly on the visible point cloud. We visualize sampled starting points and target points in Fig. 8. We use inverse kinematics to move the robot end-effector in a straight line from start to end.

### B.2   SBML structure generation module

**VSF particle set generation**   We first segment out the object using a rough bounding box relative to the fixed base. We define a volumetric grid on this bounding box and do raycasting from the camera point to sample a set of points in the occluded region. The grid cells have size ~1cm for artificial plants and ~0.5cm for shoes. We visualize this procedure in Fig. 9. We downsample the points sampled from the grid to $10,000 \sim 30,000$ particles depending on the size of the object.

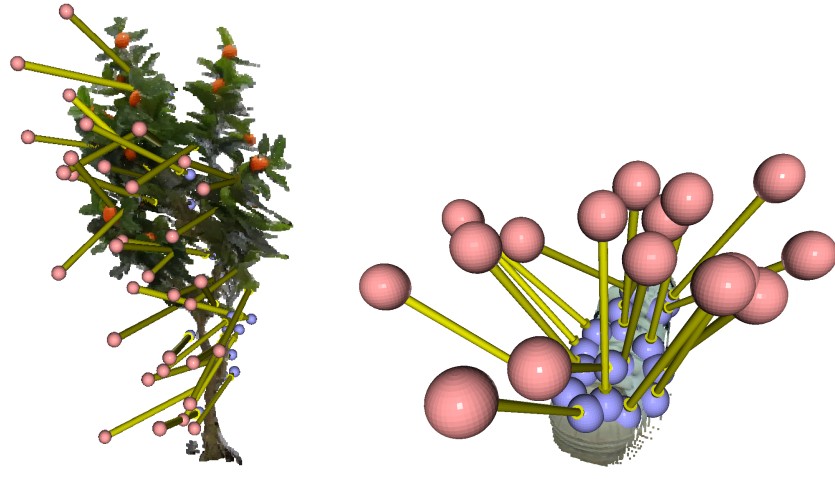

Figure 8: Sampled touches on an orange tree and a white sneaker. Pink balls indicate the starting points and blue balls indicate the target points. Yellow arrows indicate the straight line from start to end.

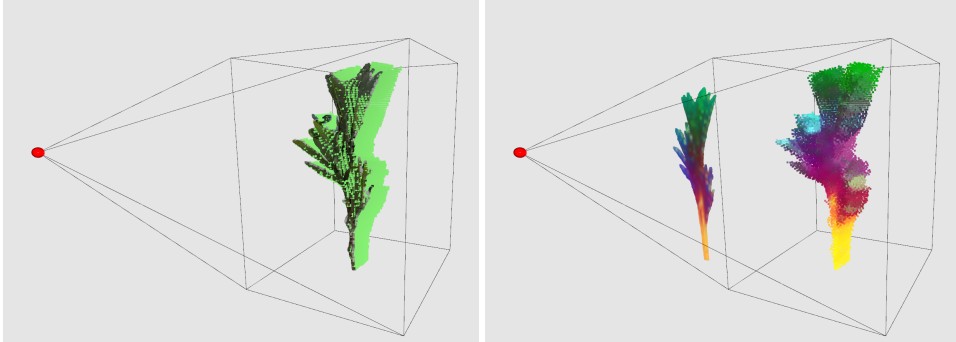

Figure 9: Left: VSF particle sampling for an artificial plant. The red ball indicates the camera point. The green points are sampled VSF points. The bounding box is defined as large enough to enclose the object. Right: Project DINOv2 features on the image plane to 3D points, colors are from the first three dimensions in PCA of the features.

**Visual features** We use DINOv2 [46] to generate dense visual features $v_\alpha$ in our experiments. We use pre-trained weights available on their website. By default, we use the small-version model that creates a 384-dimensional feature per 14x14 image patch. For visible points in VSF, we project their 3D coordinates to the image plane and use linear interpolation to compute the feature. For occluded points in VSF, we use the average features of its 10-closest visible points. Therefore particle $i$ has a feature $v_{\alpha,i}$. Also, we normalize the visual features using statistics from the training and validation dataset to speed up learning.

### B.3 SBML meta-prior architecture

For heterogeneous components, we choose a 4-layer MLP with dimensions $[384, 256, 128, 2]$. The 2-D output is the Gaussian distribution's mean and variance, and the ReLU operation in the last layer ensures non-negative output. The meta-parameters $\psi_{HE}$ are the weights of this neural network.

The meta-prior for segmented and homogeneous components uses a linear mean model and a learnable variance. The meta-parameters $\psi_{SEG}$ or $\psi_{HO}$ are linear weights and variance. We choose this low-capacity model because of the small number of segments and tasks in the meta-training dataset.

We initialize the model uncertainty by setting an initial bias for variance output in MLP and initial variance value in the linear model.

### B.4 iMAML implementation

We use an MLP to map robot action to tactile reading in iMAML implementation. For the *no-vision* version, the network input is the 7-d joint angle, and we have 6 hidden layers with dimensions [32, 128, 64, 32, 16]. The *with vision* version takes an additional 384-dimensional DINOv2 classification token to distinguish different objects. The 384-dimensional features are encoded using a [64, 7] MLP and the output is concatenated with the 7-d joint angle as 14-d network input. The output layer is 7-d to predict joint torques for the *plant* dataset and 1-d to predict pressure reading for the *shoe* dataset.

We used gradient descent for the inner loop iMAML adaptation. For both benchmarks, we use 2.0 for the regularization strength, 0.001 as the outer loop learning rate, and a total of 400 outer epochs. For the conjugate algorithm, we use 100.0 for the conjugate gradient damping and perform 10 steps. The inner loop learning rate was set 0.001 for *shoe* and 0.01 for *plant*. A total of 30 inner loop gradient descent steps are taken for both benchmarks.

## C Supplementary experiment results

### C.1 Additional baseline results

We provide results with alternative baseline implementations in Tab. 4 and 5. For the non-structured baseline learned with iMAML, we consider a no-vision baseline that does not use visual information. We do not see a significant performance difference compared to iMAML with vision in Tab. 1 and Tab. 2. This indicates that the visual features cannot enhance the meta-learning of tactile responses without structure in such a real-world dataset. For the Gaussian prior baseline, since the average mean tends to overestimate the stiffness, we also consider a second variant (zero mean) that simply sets the mean to zero. This zero Gaussian prior has a better zero-shot prediction on the Plant benchmark, but its 10-shot error is even larger than the 0-shot error of SBML prior.

Table 4: Joint torque prediction error (Nm) ± std over query set on Plant benchmark.

|  | iMAML no-vision (broad dataset) | Gaussian prior (zero) |
|---|---|---|
| 0-shot | $3.95 \pm 0.85$ | $3.55 \pm 0.00$ |
| 1-shot | $5.71 \pm 3.52$ | $3.43 \pm 0.03$ |
| 5-shot | $5.10 \pm 2.16$ | $3.09 \pm 0.04$ |
| 10-shot | $4.40 \pm 0.93$ | $2.88 \pm 0.03$ |
| VSF all-shot 2.57 | | |

Table 5: Tactile prediction error (hPa) ± std over query set on Shoe benchmark.

|  | iMAML no-vision (broad dataset) | Gaussian prior (zero) |
|---|---|---|
| 0-shot | $7.44 \pm 1.93$ | $9.21 \pm 0.00$ |
| 1-shot | $7.22 \pm 2.77$ | $8.08 \pm 0.14$ |
| 5-shot | $6.38 \pm 1.23$ | $5.65 \pm 0.22$ |
| 10-shot | $6.22 \pm 0.89$ | $4.63 \pm 0.09$ |
| VSF all-shot 4.10 | | |

### C.2 Uncertainty of SBML prior

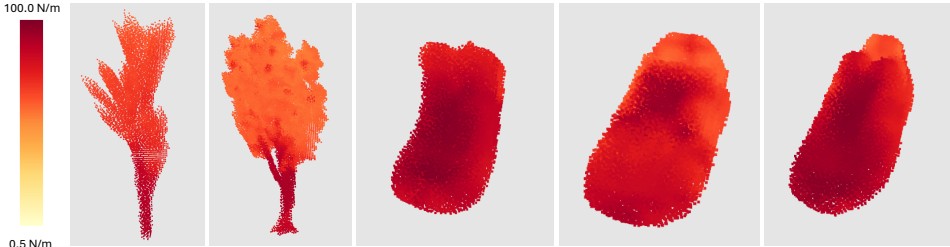

Figure 10: SBML prior standard deviation in stiffness. The color map is in the log scale.

The SBML prior also learns the uncertainty of stiffness in VSF model. We visualize the standard deviation of learned Gaussian prior in Fig. 10. For plants, we can see the stiff branch region is predicted with high uncertainty, likely due to high variation in branch stiffness caused by detailed geometric features like thickness and distance to the plant base. On the other hand, shoes' stiffness uncertainty is overall higher than plants', probably caused by the more significant material variation in the shoe dataset.

### C.3 Unified prior over plants and shoes

To evaluate the capacity of our method, we test whether it can learn a unified VSF prior between the plant and shoe categories. We learn a **single** prior using both plant and shoe broad datasets with the same training configuration. We visualize the unified prior mean prediction in Fig. 12. We qualitatively find that the unified prior in the last row has a prediction similar to the prior trained on data from individual category datasets in the second row. Quantitative results are given in Table 6 and Table 7. Errors are comparable to the Plant-only and Shoe-only datasets, demonstrating that SBML can learn multi-category visual-tactile models without significant change in performance.

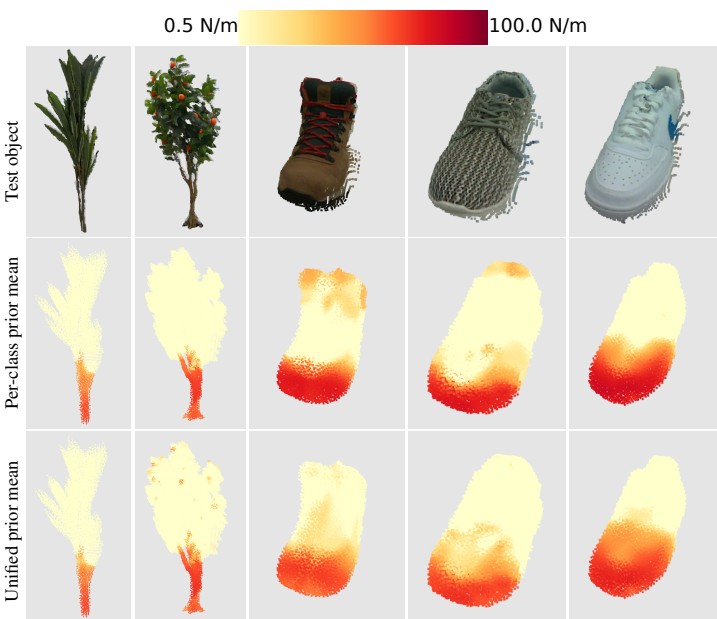

Figure 11: Here we compare the zero-shot prediction using a **single** prior learned from both plant and shoe dataset data (last row) with zero-shot prediction using data of their specific class (second row). Here per-class means testing on plants using prior trained on broad plant dataset and testing on shoes using prior trained on broad shoe dataset.

Table 6: Joint torque prediction error (Nm) ± std over query set on Plant benchmark using per-class prior or unified prior.

|  | Plant data only (broad dataset) | Shoe + Plant data (broad dataset) |
|---|---|---|
| 0-shot | 2.80 ± 0.05 | 2.95 ± 0.07 |
| 1-shot | 2.76 ± 0.05 | 2.92 ± 0.07 |
| 5-shot | 2.67 ± 0.05 | 2.83 ± 0.06 |
| 10-shot | 2.61 ± 0.04 | 2.77 ± 0.05 |
| VSF all-shot 2.57 | | |

Table 7: Tactile prediction error (hPa) ± std over query set on Shoe benchmark using per-class prior or unified prior.

|  | Shoe data only (broad dataset) | Shoe + Plant data (broad dataset) |
|---|---|---|
| 0-shot | 7.43 ± 0.25 | 7.05 ± 0.49 |
| 1-shot | 6.77 ± 0.26 | 6.52 ± 0.41 |
| 5-shot | 5.05 ± 0.20 | 5.06 ± 0.20 |
| 10-shot | 4.18 ± 0.04 | 4.32 ± 0.06 |
| VSF all-shot 4.10 | | |

## C.4 Per-point stiffness estimation

In this section, we provide a better qualitative stiffness estimation using per-point touches. Note this estimation does not use the VSF model and is only for qualitative comparison. We setup another robotic system using the UR5 arm's built-in force/torque sensor to measure contact forces when the object is touched with a conical probe. Touched points are selected by voxel-downsampling the surface point cloud to 1 cm, and the probe is depressed normal to the surface and stopped when a force threshold is reached. The per-point stiffness value is estimated by dividing the maximum contact force over the displacement. For better visualization, we interpolated estimated stiffness on a triangle mesh of the object. The learned SBML prior qualitatively matches trends in the per-point estimation, with the toe region estimated to be more stiff than the tongue region.

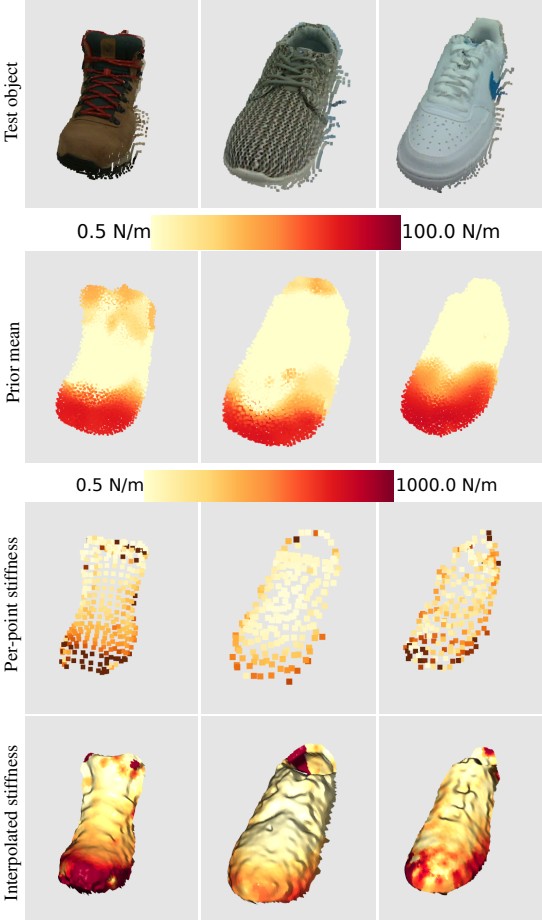

Figure 12: Comparing VSF prior with per-point stiffness estimation, with a point probe moving normally to the object surface. An interpolated mesh is also shown. Per-point stiffness is sampled at a different density compared to VSF, and the estimated stiffness is plotted on different scales.

## C.5 Material composition experiments

The experiments of Sec. 4.5 are defined in more detail as follows. We consider three testing conditions for the composition structure $\mathcal{C}_\alpha$ : heterogeneous, heterogeneous + segment, and heterogeneous + homogeneous.

- Heterogeneous: contains $m_\alpha$ components and each component contains one spring in VSF.
- Heterogeneous + homogeneous: contains $m_\alpha + 1$ components, where $m_\alpha$ components contain one spring with stiffness $K_{i,HE}$ and the last component assigns a uniform stiffness across all the springs $K_{HO}$. Following Section III.C, the stiffness of each spring is thus $K_i = K_{HO} + K_{i,HE}$.

- Heterogeneous + segment: contains $m_\alpha + n_s$ components, where $m_\alpha$ components contain one spring and the $n_s$ components contain the springs in one segment. We use 10 segments generated by GMM clustering on DINOv2 features.

## C.6 Effects of visual features and prior structure

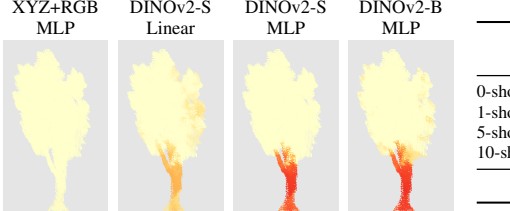

| | XYZ+RGB MLP | DINOv2-S Linear | DINOv2-S MLP | DINOv2-B MLP |
|---|---|---|---|---|
| 0-shot | $3.84 \pm 0.11$ | $4.30 \pm 0.15$ | $2.80 \pm 0.05$ | $2.75 \pm 0.08$ |
| 1-shot | $3.77 \pm 0.12$ | $4.19 \pm 0.15$ | $2.76 \pm 0.05$ | $2.74 \pm 0.09$ |
| 5-shot | $3.51 \pm 0.04$ | $3.81 \pm 0.10$ | $2.67 \pm 0.05$ | $2.69 \pm 0.06$ |
| 10-shot | $3.27 \pm 0.09$ | $3.50 \pm 0.11$ | $2.61 \pm 0.04$ | $2.64 \pm 0.05$ |
| | VSF all-shot 2.57 | | | |

Table 8: Left: Effects of the different visual features and prior structures on the meta-learning dataset. We visualize the zero-shot prior mean of the orange tree in the test set. Right: Joint torques prediction error (Nm) on Plants dataset, for SBML with different visual features and prior.

We evaluate how the choice of visual feature and prior architecture affect meta-learning performance. To avoid meta-overfitting, our mitigation strategies (in Sec 3.3 the last paragraph) help, but do not eliminate the risk. Since our dataset is relatively small, the feature relevance and prior capacity must be chosen carefully.

We explore combinations of the following:

- Visual features: we use normalized XYZ coordinates + RGB colors as naive visual features. We also compare DINOv2-small (DINOv2-S) which outputs a 384-dimensional feature vector and DINOv2-base (DINOv2-B) which outputs a 768-dimensional feature.
- Prior structures: we consider using linear or MLP prior structures. The linear prior assumes homoscedastic uncertainty independent of input.

Tab. 8 shows learned prior with different visual features and the quantitative results on the broad plant dataset. The smaller capacity models, XYZ+RGB / MLP and DINOv2-S / linear, underfit the data. The larger capacity models (DINOv2-S / MLP and DINOv2-B / MPL) are more expressive and can capture meaningful patterns. We chose DINOv2-S / MLP as our default model to avoid overfitting in smaller training datasets.

We also observe failure cases where DINOv2 features generalize poorly. One example is the *slipper* class as shown in Fig. 9. Even though the learned prior seems to capture that the toe is stiffer than the upper (i.e., the term for the part that fits over the foot), the upper's stiffness is underestimated. Hence, the prediction error is not significantly better than a naive Gaussian prior. We hypothesize that the diverse stiffness distribution of slippers might cause the failure. Here, the stiffness depends on the thickness of the upper part and other fine geometry details that are not captured in the image.

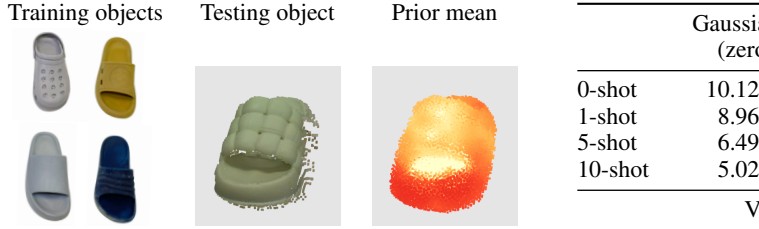

| | Gaussian prior (zero mean) | SBML prior |
|---|---|---|
| 0-shot | $10.12 \pm 0.00$ | $9.30 \pm 0.35$ |
| 1-shot | $8.96 \pm 0.46$ | $8.43 \pm 0.63$ |
| 5-shot | $6.49 \pm 0.26$ | $6.42 \pm 0.29$ |
| 10-shot | $5.02 \pm 0.11$ | $5.12 \pm 0.15$ |
| | VSF all-shot 4.55 | |

Table 9: Slipper failure case. Left: the prior poorly generalizes from meta-training objects. Right: SBML improves tactile prediction errors (hPa) marginally compared to using an uninformed prior.

## C.7 Effects of training dataset size

We also evaluate the SBML prior's performance given different sizes of the training dataset. We consider reducing the dataset size by randomly subsampling the objects or subsampling touches-per-object. Results are shown in Tab. 10 and Tab.11. As expected, as the dataset shrinks, performance tends to degrade and have a larger variance. We also observe that for *shoes* it is more important to have more objects and fewer touches per object, where using 40% touching sequences and all objects can achieve good performance. Because the *shoes* has only about 20–30 touches per object, 10% touch subsampling is extremely sparse and results are noisy. Also, it appears that subsampling at 70% touches for *shoes* gives optimal performance, but the results are not statistically significant.

Table 10: SBML zero-shot joint torques prediction error (Nm) on Plants dataset with different training data.

|          | 10% seq | 40% seq | 70% seq | 100% seq |
|----------|---------|---------|---------|----------|
| 10% obj  | $4.97 \pm 2.17$ | $4.24 \pm 2.09$ | $3.94 \pm 1.13$ | $3.41 \pm 0.49$ |
| 40% obj  | $4.07 \pm 1.48$ | $3.04 \pm 0.19$ | $2.85 \pm 0.21$ | $3.27 \pm 0.66$ |
| 70% obj  | $3.66 \pm 1.24$ | $3.24 \pm 0.27$ | $2.82 \pm 0.17$ | $2.83 \pm 0.14$ |
| 100% obj | $3.18 \pm 0.36$ | $3.33 \pm 0.29$ | $2.82 \pm 0.07$ | $2.80 \pm 0.05$ |

Table 11: SBML zero-shot tactile prediction error (hPa) on Shoes dataset with different training data.

|          | 10% seq | 40% seq | 70% seq | 100% seq |
|----------|---------|---------|---------|----------|
| 10% obj  | $8.84 \pm 1.88$ | $8.49 \pm 1.90$ | $8.10 \pm 0.99$ | $8.38 \pm 1.10$ |
| 40% obj  | $9.68 \pm 2.07$ | $8.87 \pm 2.26$ | $8.82 \pm 1.41$ | $8.76 \pm 0.92$ |
| 70% obj  | $12.17 \pm 2.81$ | $9.07 \pm 2.26$ | $8.17 \pm 2.42$ | $8.44 \pm 1.45$ |
| 100% obj | $9.70 \pm 1.57$ | $7.50 \pm 0.56$ | $7.03 \pm 0.63$ | $7.43 \pm 0.26$ |

## C.8 Evaluation with different viewpoints

Finally, we evaluate whether the learned prior is robust to different viewpoints. We found that DINOv2 visual features are robust to viewpoint variation and SBML predicts stiffness with semantic correspondences. We rotate a shoe left and right by $45$ degrees and use the SBML prior trained on the broad shoe dataset to make predictions. As shown in Fig. 12, the learned prior predicts the semantic sole part with a high stiffness between different viewpoints.

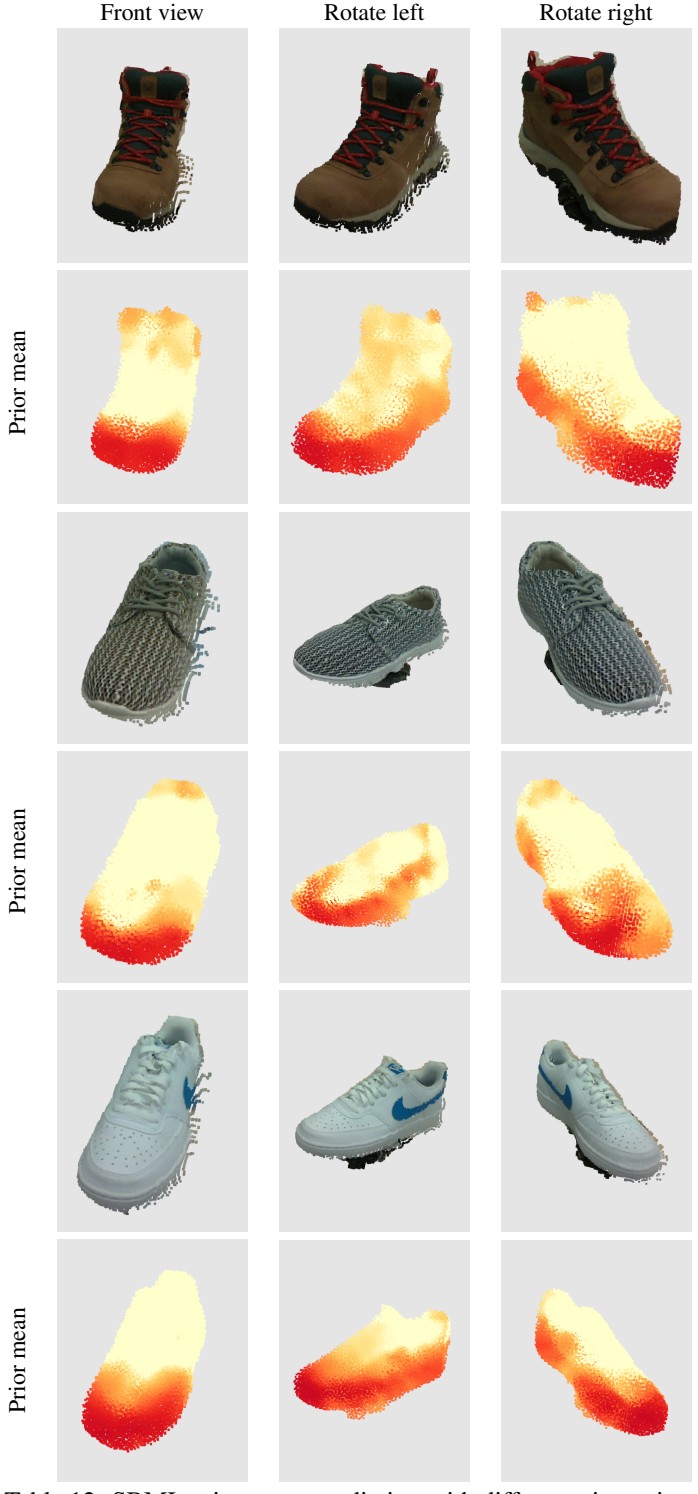

Table 12: SBML prior mean prediction with different viewpoints.

