# OpenReview forum: "Structured Bayesian Meta-Learning for Data-Efficient Visual-Tactile Model Estimation"
_robot-learning.org/CoRL/2024/Conference — CoRL 2024_

### Official Review · Reviewer_HDBy · 2024-07-15
**Hierarchical Bayesian Learning for Soft Object Mapping. Good approach but requires clarification.**

**Originality:** 4
**Technical Quality:** 4
**Clarity Of Presentation:** 4
**Potential Impact:** 3
**Recommendation:** 4
**Confidence:** 4

**Review:**

The paper tackles a relevant and rather under explored problem: inferring the compliance of objects over their surface area. The problem is well motivated. I like the discussion of differences between tactile and vision in the introduction. It is sensible to formulate the problem as one of hierarchical Bayesian inference as the authors do. The need for priors is well motivated with the unavoidable sparsity of touch data.
The method as explained seems sensible, however there are a few open questions regarding the design of the method and also given the results that need to be clarified and potentially require additional work beyond writing. Please see these issues below.
Beyond these issues I think it would greatly benefit the work if the authors actually went further and tested transfer between categories, as they mention in the limitation section. (I want to mention this here and don't include it in the list below because I don't think it necessarily needs to be addressed. However, it would be nice).

**Quality Of The Limitations Section:**

2

**Questions For Rebuttal:**

- L214: why are these different? Please motivate.

- As far as I can see the objects are always viewed from the same perspective -- both, shoes and plants. If this is true the results we see could strongly overfit to the perspective. It is necessary to provide experimental data that shows the method does not just learn "further down in the image means more stiff". The objects could be presented in random rotations or at least be approached from different directions while in their "natural orientation" (sole/stem downwards).

- L239: It is not a good idea to use the output of your own method as "proxy for ground truth" -- especially because the results in Figure 5 do look rather erroneous. I don't believe the middle part of the brown boot is as soft as the middle part in the other two shoes. The brown boot should be generally way more stiff, I suppose. Shouldn't it be possible to collect ground truth by somewhat exhaustively mapping the objects square centimeter by square centimeter? Please provide a sensible ground truth, or at least explicitly acknowledge that you don't quantitatively assess the quality of the method. Sensible ground truth would be the way stronger option here though.

- L282: Table 1 itself doesn't state it would be zero shot performance, but the text in this line implies so. Something is off here. Table says many-shot.

- L146, L149: this enumeration is unexpected. what are you enumerating here?

- L156-L178: This is confusing and also comes surprisingly as we did not talk about components up until this point. Please make sure that there is enough meta description before this point so that we understand how this fits into the big picture. This is also missing from the overview schematics for the method.

**Robotics Focus:**

4

**Summary Of Paper:**

This paper presents a method for visuo-tactile mapping of objects. It employs a hierarchical Bayesian approach, where the mapping problem is factorized into learning prior knowledge about object classes and then using this object-class specific prior knowledge in the visuo-tactile mapping of individual object instances. The paper is motivated by the problem to predict the tactile behavior (stiffness) over full object surfaces, but from the sparse data collected using only few tactile interactions with the same object. The hierarchical Bayesian approach tackles this problem as it learns useful prior information that allows the framework to predict the stiffness of object patches either using only vision (zero-shot), or by extrapolating tactile information from other interaction points on the instance.

**Summary Of Recommendation:**

Relevant problem tackled by an approach that fits (hierarchical Bayesian approach). However the authors should provide some experiments from other perspectives to make sure the method doesn't overfit and should also ideally compare against real ground truth (and not against outputs of their own algorithm)..

---

### Official Review · Reviewer_oWYD · 2024-07-20
**Meta-Learning with Multisensory Data; Stiffness Estimation**

**Originality:** 2
**Technical Quality:** 3
**Clarity Of Presentation:** 3
**Potential Impact:** 2
**Recommendation:** 2
**Confidence:** 3

**Review:**

The paper is well-written and clear to read. The graphs and illustrations are good. The idea and method are well-explained.

Strengths:
- The methods is novel.
- Sufficient robot demonstrations on two different datasets
- Data efficiency is good. The model can do a accurate estimation with zero-shot visual prediction and few-shot adaptation with tactile.

Weakness:
- My main concern is the generalization of this method.
- As the authors point out in limitations, this approach cannot generalize cross object categories, and even in the shoe dataset, since the shoe appearance and material are quite different from each other, thus the performance is worse across shoes.
- The two datasets are all not large. I can imagine that even if the model is trained on tree dataset and then fine-tuned on shoe dataset, it will probably "forget" the features it learned in tree datasets and perform bad on trees. Perhaps large models can be leveraged to identify object categories beforehand to enable better generalization.

**Quality Of The Limitations Section:**

1

**Questions For Rebuttal:**

As mentioned in above review, I'm interested in more results cross object categories, even if they are not deformable. If you use some vision-based tactile sensors, it is also possible to retrieve surface stiffness from the readings. Have you tried any large multimodal datasets such as Objectfolder benchmark [1], or Touch and Go [2]? I think Objectfolder benchmark contains visual and tactile data and material properties that might be able to infer stiffness. Touch and Go has data collected in the wild from different surfaces.

[1] Gao, Ruohan, et al. "The objectfolder benchmark: Multisensory learning with neural and real objects." Proceedings of the IEEE/CVF Conference on Computer Vision and Pattern Recognition. 2023.
[2] Yang, Fengyu, et al. "Touch and go: Learning from human-collected vision and touch." arXiv preprint arXiv:2211.12498 (2022).

**Robotics Focus:**

4

**Summary Of Paper:**

The authors present a method Structured Bayesian Meta-Learning to use visual-tactile data to estimate stiffness of deformable objects. The method leverages Bayesian meta-learning with an incorporated structure generation module, allowing it to adapt to objects with different parameter spaces. The approach does a zero-shot vision prediction first and then few-shot adaptation using tactile signals.

**Summary Of Recommendation:**

My main concern is the generalization of the method.

---

### Official Review · Reviewer_57dj · 2024-07-21
**Quick Overview of Structured Bayesian Meta-Learning: A Data-Efficient Approach for Visual-Tactile Model Estimation**

**Originality:** 4
**Technical Quality:** 4
**Clarity Of Presentation:** 4
**Potential Impact:** 2
**Recommendation:** 3
**Confidence:** 3

**Review:**

### Quality
The quality of this paper is high, with robust empirical validation and a well-structured methodology. The authors provide a comprehensive explanation of the proposed Structured Bayesian Meta-Learning (SBML) approach, detailing its components and the rationale behind its design. The experimental setup and results are clearly presented, showcasing the effectiveness of SBML in comparison to existing methods.

### Clarity
The paper is generally clear and well-written.

### Strengths:
1.  The introduction of SBML addresses a critical challenge in visual-tactile model estimation, offering a novel solution that leverages Bayesian meta-learning.
2.  The method's capability for zero-shot and few-shot learning is particularly valuable in practical applications with limited data.

### Weaknesses:
1.  The evaluation is currently limited to the VSF model, which only predicts tactile response.
2.  The use of a Gaussian prior assumes a single mode of material parameters conditioned on visual features, which may not capture more complex, multi-modal distributions effectively.

**Quality Of The Limitations Section:**

2

**Questions For Rebuttal:**

1. Can the authors provide additional experiments or validation results using other deformable object models, such as Finite Element Method (FEM) or Graph Neural Dynamics Models, to demonstrate the broader applicability of SBML?

2. The paper mentions that tactile data is often sparse and noisy. Can the authors elaborate on the specific techniques used to mitigate these issues in the SBML framework?

**Robotics Focus:**

4

**Summary Of Paper:**

The paper addresses the challenge of estimating visual-tactile models for deformable objects, where vision is hindered by occlusion and touch data is sparse and noisy. The proposed approach leverages Bayesian Meta-Learning (BML) to create a data-efficient method for model estimation that can generalize from diverse training objects. The paper introduces Structured Bayesian Meta-Learning (SBML) to deal with the issue of different parameter spaces for various visual-tactile models. SBML incorporates heterogeneous physics models to learn from objects with varying appearances and geometries, enabling both zero-shot vision-only prediction and few-shot adaptation with tactile data. The experiments validate the efficacy of SBML on heterogeneous objects, namely plants and shoes, demonstrating superior performance in force and torque prediction accuracy in both zero- and few-shot settings.

**Summary Of Recommendation:**

Overall, "Structured Bayesian Meta-Learning for Data-Efficient Visual-Tactile Model Estimation" is a high-quality, original, and significant contribution to the field. The proposed SBML approach addresses key challenges in visual-tactile model estimation, offering a data-efficient solution that generalizes from diverse training objects. While the method shows strong potential, future work should aim to extend its scope to other models and improve its capability to handle multi-modal distributions and generalize across different object categories.

---

### Author Rebuttal · Authors · 2024-08-08

The zip file contains revised_manuscript.pdf and new_supplement_experiments.pdf

To assist the reviewer in finding new experiment results, the new_supplement_experiments.pdf highlights new results we added in the appendix, they are identical to Appendix C.6-8 (pages 21-23) in the main revised manuscript.

---

### Decision · Program_Chairs · 2024-09-04

**Decision:**

Accept

**Comment:**

The reviewers found the paper to be well written with a clear and signficiant contribution. There were some concerns regarding generalization and evaluations. The rebuttal helped to strengthen the paper and increase its potential impact. The paper seems well suited for CoRL.